# Sensitive Skins May Be Neuropathic Disorders: Lessons from Studies on Skin and Other Organs

**Laurent Misery** 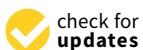

Department of Dermatology, University Hospital of Brest, 29200 Brest, France; laurent.misery@chu-brest.fr

**Abstract:** Sensitive skin can be considered a neuropathic disorder. Sensory disorders and the decrease in intra-epidermal nerve ending density are strong arguments for small-fiber neuropathies. Sensitive skin is frequently associated with irritable bowel syndrome or sensitive eyes, which are also considered neuropathic disorders. Consequently, in vitro co-cultures of skin and neurons are adequate models for sensitive skin.

**Keywords:** sensitive skin; itch; pain; neuron; neuropathic; irritable bowel syndrome; dry eye disease

## 1. Introduction

To the best of my knowledge [1], the first report on sensitive skin was made in 1947 [2]. However, sensitive skin only became an area of interest in the 1980s [3,4]. Sensitive skin was described as a syndrome in 2006 [5], and a consensus definition was published in 2017 [6].

Using the Delphi method, the special interest group on sensitive skin of the International Forum for the Study of Itch (IFSI) defined sensitive skin as follows: "a syndrome defined by the occurrence of unpleasant sensations (stinging, burning, pain, pruritus, and tingling sensations) in response to stimuli that normally should not provoke such sensations. These unpleasant sensations cannot be explained by lesions attributable to any skin disease. The skin can appear normal or be accompanied by erythema. Sensitive skin can affect all body locations, especially the face" [6].

A meta-analysis, including 13 studies, showed that cosmetics and physical (variations in temperature, cold, heat, wind, sun, air conditioning, wet air and dry air), chemical (water and pollution) or psychological (emotional) factors were associated with sensitive skin. The most important factors were cosmetics (odds ratio (OR) = 7.12 [3.98−12.72]), wet air (OR 3.83 [2.48−5.91]), air conditioning (OR 3.60 [2.11−6.14]) heat (OR 3.5 [2.69−4.63]) and water (OR 3.46 [2.56−4.77]) [7].

Following a first epidemiological study in the United Kingdom in 2001 [8], numerous studies have been conducted in many countries throughout the world, including Belgium, France, Germany, Greece, Italy, Portugal, Spain, Switzerland, the United States, Brazil, Japan, Russia, Korea, China and India [9]. The global prevalence of "sensitive skin" is approximately 50%, with variations among some countries [9]. A comparison of four studies in the USA suggests that the frequency of sensitive skin might increase from 50 to 85% [10] over time, while comparisons of studies on the French population show a smaller increase [11].

The IFSI special interest group on sensitive skin published a position paper on the pathophysiology and management of sensitive skin [12]. A multifactorial origin was suggested after the discussion of many putative mechanisms. However, it was concluded that sensitive skin is not an immunological disorder but is related to alterations of the skin nervous system, with which skin barrier abnormalities are frequently associated, but without any direct relationship. According to the high frequency of sensitive skin, a single pathophysiological mechanism is debatable. However, a growing body of data supports

the hypothesis that sensitive skin is a neuropathic disorder [13,14]. The aim of this review is to identify all these data.

## 2. Arguments for Sensitive Skins as Small-Fiber Neuropathies

Small fiber neuropathies (SFNs) are disorders of unmyelinated C-fibers and poorly myelinated A-delta fibers, which induce pruritus and other cutaneous paresthesia [15–17]. A recent systematic literature review showed an unmet need of broadly accepted diagnostic criteria, although the most common set of mandatory criteria to diagnose were sensory symptoms (60% of studies), pain (19% of studies), small fiber signs (20% of studies), absence of large fiber signs (62% of studies), reduced intra-epidermal nerve fiber density (IENFD) (38% of studies), and autonomic symptoms (1% of studies) [18]. Nonetheless, the joint task force of the European Federation of Neurological Societies (EFNS) and the Peripheral Nerve Society (PNS) agreed to consider that skin biopsy with the quantification of the IENFD, using generally agreed upon counting rules, is a reliable and efficient technique to assess the diagnosis of SFN [19].

Thanks to a large immunohistochemical study, we tested all pathophysiological hypotheses for sensitive skin and we evidenced only arguments for the neuronal hypothesis [20]: the IENFD was significantly reduced in patients with sensitive skin by comparison with controls, which means that the Aδ or C fibre population was altered. The CGRP-immunoreactive nerve fibre density was also reduced in these skins.

In another study, performed on 70 women with sensitive skin, we previously found that 20% exhibited characteristics of neuropathic pain based on an evaluation using the DN-4 (*Douleur Neuropathique*-4) questionnaire [21].

In a case-control study, we showed that both the DN-4 and Neuropathic Pain Symptom Inventory (NPSI) scores were significantly increased in patients with sensitive skin compared with the control group [22]. Using quantitative sensory testing (QST), this study revealed a significant decrease in the heat-pain threshold in the sensitive skin group versus the control group, demonstrating that sensitive skin is associated with an alteration of C fibres [23–25]. No difference was found in the vibration detection threshold or the cold detection threshold, which implies the absence of damage to other skin fibres, such as Aβ and Aδ fibres [23–25].

All together, these findings are similar to all criteria for SFN diagnosis [15,26]: (1) clinical signs of small-fibre impairment (pinprick and thermal sensory loss or allodynia or hyperalgesia, or any combination of the three) for whichever distribution is consistent with peripheral neuropathy (i.e., length-dependent or non-length-dependent neuropathy); (2) abnormal warm or cold threshold or both as assessed using QST; (3) reduced IENFD [26].

Hence, we are allowed to consider sensitive skin as a minor equivalent of SFNs, with alterations in cutaneous small nerve fibres, especially in unmyelinated C fibres. These alterations, inducing neuropathic pain and a decrease in heat-pain threshold detection, suggest the hyper-reactivity of nerve endings in response to environmental factors.

## 3. Lessons from the Bowel and the Eyes

We have shown that sensitive skin is frequently associated with irritable bowel syndrome (IBS) [27] or sensitive eyes [28], which should be also considered as SFNs.

IBS is characterized by abdominal pain or discomfort in association with bloating and/or defecation disorders and/or altered bowel habits [29]. Hypersensitivity to visceral stimuli and reduction in pain or discomfort thresholds in these patients are considered major clinical features of IBS [30]. The pathophysiology of such visceral hypersensitivity is likely multifactorial, involving both peripheral and central neural mechanisms, based on measurements of subjective pain thresholds evoked by experimental visceral stimuli [31], as well as objective data demonstrating the hyperexcitability of pain systems in IBS patients [32]. Epidemiological studies demonstrated that IBS is frequently associated with several comorbidities, both intestinal and extra-intestinal, especially in the context of chronic overlapping pain conditions [33].

We demonstrated that there is an association between IBS and sensitive skin, which are two painful and frequent conditions found especially in women [27]. Sensitive skin was more frequent in people with IBS. Reciprocally, IBS was more frequent in people with sensitive skin and associated with its severity. Recent advances in molecular neurophysiology provide knowledge to better understand the underlying mechanism in pain generation in IBS patients. The sensitization of peripheral nociceptive afferents, more specifically high-threshold afferents, has been proposed as one of the principle mechanisms in the development of visceral hypersensitivity [34]. In addition to direct neuronal activation, low concentrations of proteases, histamine, and serotonin can chronically sensitize nociceptors, such as TRP channels, leading to persistent aberrant pain perception [35]. Neurogenic inflammation is strongly involved in the pathogenesis of both IBS [36] and SSS [37].

We also showed that sensitive eyes are reported by half of the population, with a higher frequency in women [28]. The subjects who reported sensitive eyes were more numerous, according to the severity of skin sensitivity. More than half of the subjects with sensitive eyes thought that they were sensitive to sun exposure, dust, computer or touch pads or dry air. The concept of sensitive eyes is not commonly used by ophthalmologists and remains to be more precisely defined. Because the cornea/ocular surface is the equivalent of skin, the existence of sensitive corneas should be defined in a similar manner. Cornea is the most densely innervated tissue [38]. The Tear Film & Ocular Surface Society (TFOS) Dry Eye Workshop (DEWS) II pathophysiology report introduced the notion of neuropathic pain to describe all clinical pictures where the patient complains of pain of any degree that is associated with no obvious lesion [39]. Ophtalmologists use the term "dry eye disease" (DED) to name the most frequent ocular surface disease, ranging from 5 to 50% of the world population [40,41] and its new definition includes neurosensory abnormalities [42].

Similar to epidermal innervation, corneal innervation consists of many C fibres that are equipped with a large variety of sensor proteins, such as transient receptor proteins (TRPs) [43,44], which allow for fine perceptions from nociceptors or pruriceptors. Sensitive skin is related to neurogenic inflammation and peripheral sensitization to pain [14]. Peripheral sensitization is due to the hyperactivation of A-delta and C nociceptive fibres by various factors, which is accompanied by a barrage of impulses that activate postsynaptic receptors, resulting in the escalation of the hyperexcitability of secondary neurons with the marked amplification of peripheral stimuli [45]. Such phenomena have been largely described in the cornea [46] and are probably highly involved in sensitive corneas. Many morphological and functional changes of corneal nerve terminals were observed in DED patients, as well as peripheral and central neuroimmune interactions in the development of corneal hypersensitivity or cellular and molecular changes of corneal neurons [47,48]. These results should inspire research on sensitive skin.

Peripheral and central sensitization to pain and itch are possible in sensitive skin, sensitive eyes and irritable bowel syndrome [47]. Hence, the analysis of cerebral responses to cutaneous provocation tests in self-perceived sensitive and non-sensitive skin subjects using functional magnetic resonance imaging (fMRI) [49] showed that cerebral activity was significantly increased in the sensitive skin group. In sensitive skin, activity extended only into the ipsilateral primary sensorimotor cortex and the bilateral peri-insular secondary somatosensory area. These findings suggest that, compared with control subjects, subjects with self-perceived sensitive skin exhibit specific cerebral activation during skin irritation tests, which might be related to a central sensitization to itch and pain, in addition to the peripheral sensitization as an SFN.

Other organs could be involved in similar disorders to sensitive skin. Especially, the triggering of cough by relatively innocuous stimuli suggests heightened sensitivity of the sensory nerve pathways alluded to earlier that normally serve to detect and respond to harmful airway irritants. In these circumstances, the chronic cough should not be considered as a symptom but rather as a disease entity caused by a disordered nervous system in which the concept of hypersensitivity cough is emerging [50].

## 4. Arguments for Neuro-Keratinocytic Interactions

Subjects with sensitive skin may have dry, mixed, oily or otherwise normal skin [51,52]. A systematic review of the literature showed that the levels of epidermal pH, sebum production and skin hydration were inconsistent [53]. Consequently, dry skin and sensitive can be associated, but sensitive skin cannot be reduced to being a manifestation of dry skin, and its management does not consist of the only application of emollients [12]. Nonetheless, it does not mean that there is no role of keratinocytes in sensitive skin and there are complex interactions between neurons and keratinocytes in sensitive skin [54].

Two transcriptomic studies have been performed to compare skin samples of patients with sensitive skin and controls using DNA microarray [55] or RNA sequencing [56]. Although the small sample size could make the results debatable, the authors showed the involvement of innervation and Merkel cells in the pathophysiology of sensitive skin [57]. They also suggested keratinocytic involvement and a putative role of innate immunity [57] or adiponectin deficiency in sensitive skin [55].

Keratinocytes express diverse sensory receptors present on sensory neurons, such as receptors of the TRP family, such as TRPV1, one of the main transducers of painful heat, also involved in itch transduction or TRPV4, depicted as a heat sensor [54]. While TRPV1 and TRPV4 are expressed both by sensory neurons and keratinocytes, it has recently been demonstrated that the specific and selective activation of TRPV1 on keratinocytes is sufficient to induce pain. Similarly, the targeted activation of keratinocyte-expressed TRPV4 elicits itch and the resulting scratching behaviours [58,59]. Contrary to classical conception, the intra-epidermal nerve fibres are not the exclusive transducers of pain and itch [60]. In light of these recent advances, we can consider the putative role of epidermal keratinocytes for generating unpleasant sensations characteristic of sensitive skin before the activation of nerve endings. These findings about keratinocytes demonstrate an expanded role for epithelial cells, and beyond them of the entire epidermis that may be considered as a sensory epithelium [61]. Provided that epidermal keratinocytes contribute to abnormal sensations, the words "sensitive skin" would take on their full meaning [54].

We could summarize sensitive skin as a disorder of cutaneous small nerve fibres, especially C fibres, which are equipped with sensory neuroreceptors, such as endothelin and TRP channels. These receptors are also expressed by keratinocytes. TRP channels, which were originally described as "polymodal cellular sensors" that can be activated by various physical, chemical and thermal stimuli, are now considered "promiscuous pleiotropic molecules" because the "afferent" functions can be supplemented by "effector" roles [62]. The activation of these receptors induces the release of neuropeptides, such as substance P or CGRP, that can cause inflammation, which is termed cutaneous neurogenic inflammation (CNI) [63]. Cellular interactions induce the self-maintenance of CNI, which can promote a vicious cycle. Certain G protein-coupled receptors (GPCRs) play a prominent role in these cellular interactions and contribute to self-maintenance. Protease-activated receptors 2 and 4 (PAR-2 and PAR-4, respectively) and Mas-related G protein-coupled receptors (Mrgprs) have been implicated in the synthesis and release of neuropeptides, proteases and soluble mediators from most cutaneous cell types. In patients with sensitive skin, this uncontrolled inflammation may be favoured by an adiponectin deficiency [55].

Although only reported in mice until now, the recently discovered [64] specialized cutaneous Schwann cells with extensive processes forming a mesh-like network in the subepidermal border of the skin that conveys noxious thermal to the nerve endings might also be involved in the occurrence of sensitive skin syndrome. Glia in the skin can activate pain responses [65].

## 5. Consequences for In Vitro Models of Sensitive Skin

Because sensitive skin mechanisms can be summarized to a hyperactivation of epidermal nerve endings in close relationship with hyperactivated keratinocytes, co-cultures of skin/epidermis/keratinocytes and sensory neurons could be adequate models for in vitro studies.

We initially proposed a model including equivalents of skin, dorsal root ganglia (DRGs) and spinal cord [66]. Using a co-culture between DRG neurons and keratinocytes, we further developed a simplified model based on the measurement of the release of substance P and electrophysiological measurements [67,68]. This model allowed us to demonstrate that some products were able to inhibit CNI [69]. This co-culture model was later used to evaluate the release of calcitonin gene-related peptide (CGRP) release in the co-culture [70].

To avoid animal sacrifice, models of co-culture using neuronal cell lines (F-11 or ND7-23) were tried but cellular responses were quite disappointing [71,72]. The coculture of skin equivalents and neurons was also disappointing because it is too complex and the dermis is lacking [73].

The reinnervation of human skin explants with DRG neurons is a very satisfying model for sensitive skin [74,75]. In this model, it is possible to activate sensory neurons after the application of capsaicin on the epidermis, as showed by the modification of electric currents on patched nerve fibres and the release of neurotransmitters [76]. For the screening of putative active substances for sensitive skin, it is more convenient to use a co-culture of keratinocytes and neurons, which can be also activated by capsaicin or lactic acid [77]. This is an in vitro equivalent of the in vivo stinging test [3].

## 6. Conclusions

We could conclude that sensitive skin is a very frequent disorder, which can be considered as a minor equivalent of small-fibre neuropathies and is related to the hypersensitivity of sensory nerve endings (and keratinocytes) to environmental factors, at least in many cases. Nonetheless, the patient populations that are included in different studies may be quite heterogenous, leading to quite different results with respect to the aetiology of the phenomenon.

**Author Contributions:** L.M. wrote the whole paper. The author has read and agreed to the published version of the manuscript.

**Funding:** This research received no external funding.

**Institutional Review Board Statement:** Not applicable.

**Informed Consent Statement:** Not applicable.

**Data Availability Statement:** This paper did not report any new data.

**Conflicts of Interest:** Bayer, Beiersdorf, Bioderma, Clarins, Expanscience, Galderma, Gilbert, Johnson&Johnson, La Roche-Posay, Pierre Fabre.

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
