# Peer review of "Sensitive Skins May Be Neuropathic Disorders: Lessons from Studies on Skin and Other Organs"

_cosmetics, doi:10.3390/cosmetics8010014_

Round 1
Reviewer 1 Report
Thank you very much for the invitation to review this manuscript. Sensitive skin is a relevant topic, and this is a good discussion about the current state of research. I have four minor comments:
(1) The objective of this review should be stated at the end of the introduction. It is not entirely clear so far, what question this review adresses.
(2) The headings and the structure is difficult to follow: 1 is introduction, 2 A small-fibre neuropathy, 3 Lessions from … The heading hierarchy cannot be the same and the chapter content should be restructured (based on a clear objective).
(3) Conclusion: Words are missing in the first sentence. This is also not a conclusion but rather a repetition of what was described already. What exactly do you conclude?
(4) In vitro studies: The first sentence is a very strong claim. A model is always a model.
Author Response
Thank you for your comments
We have modified the manuscript according to your remarks:
(1) The objective of this review should be stated at the end of the introduction. It is not entirely clear so far, what question this review adresses.
(2) The headings and the structure is difficult to follow: 1 is introduction, 2 A small-fibre neuropathy, 3 Lessions from … The heading hierarchy cannot be the same and the chapter content should be restructured (based on a clear objective).
(3) Conclusion: Words are missing in the first sentence. This is also not a conclusion but rather a repetition of what was described already. What exactly do you conclude?
(4) In vitro studies: The first sentence is a very strong claim. A model is always a model.
Reviewer 2 Report
This is an interesting manuscript by one of the experts in the field on sensitive skin, where parallels are drawn between the sensitive skin phenotype and other "hypersensitivity" phenotypes such as sensitive eyes or irritable bowel syndrome.
The one general suggestion I have is to address the problem a bit more explicitly that as far as I'm aware, there is no clear-cut, universally accepted and completely objective definition or test for a "sensitive skin syndrome".
Hence, the patient populations that are included in different studies may be quite heterogenous, leading to quite different results with respect to the etiology of the phenomenon. Moreover, independent confirmation of study results by other groups is frequently lacking.
Thus, one might also phrase the title and the conclusion slightly more carefully, along the lines of "Sensitive skin might be a neuropathic disorder" or "Sensitive skin: evidence from studies on skin and other organs points to a neuropathic disorder".
Author Response
Thank you for your comments
We have modified the title and the conclusion